# Long-term Christmas Bird Counts describe neotropical urban bird diversity

**María Angela Echeverry-Galvis** [ID]**\***, **Pabla Lozano Ramírez, Juan David Amaya-Espinel** [ID]

Departamento de Ecología y Territorio, Facultad de Estudios Ambientales y Rurales, Pontificia Universidad Javeriana, Bogotá, Colombia

\* ma.echeverryg@javeriana.edu.co

**Data Availability Statement:** All minimal underlying data are within Table 2 of the Supporting Information file. Data used for the manuscript are openly available at the National

## Abstract

A significant gap in understanding the response of biodiversity to urban areas is the lack of long-term studies. Most of the information on urban birds comes from studies carried out in the northern hemisphere, and they include data that don´t exceed three years. Although short-term studies contribute to knowledge about bird community diversity and their spatial distribution in urban areas, they could be biased towards more conspicuous species. One of the few multi-temporal datasets available for birds in urban areas is the Christmas Bird Count (CBC). Using annual CBC data available between 2001 and 2018 from 21 urban and peri-urban sample sites assessed from the main cities of Colombia, we identified and analyzed long-term trends on the cumulative diversity of bird communities as well as on their spatial distribution. We estimated comparative trends in richness, number of individuals counted, similarity, and complementarity of avifauna for each city and sample site based on their responses to urbanization and dietary guilds. We identified almost a quarter of the species registered in Colombia (464 of 1954). The representativeness of the community obtained for 18 years exceeds 84%, showing richness that ranges between 214 and 278 species in the three cities. Bird species and individuals registered showed wide variation of the sample sites. We found more dwellers, insectivorous and granivorous species in urban areas, with frugivores relegated to peri-urban sites, usually coinciding with avoider species. Natural peri-urban areas and intra-urban wetlands and urban parks were the most important refuges for birds and maintained the highest avoider and utilizer species richness. Long-term inventories are fundamental for determining consolidated bird diversity and distributional patterns. This information established a baseline for decision-making and applying recommendations that allow reconciling the growing demand for urban areas with the need to preserve the native avifauna in megadiverse Neotropical countries such as Colombia.

## Introduction

The role of global urban growth in the current biodiversity crisis is a growing worldwide concern [1, 2], particularly in the developing megadiverse countries, where urban areas are increasing rapidly [3]. Urbanization usually leads to severe land-cover and land-use changes,

Audubon Society Christmas Bird count website (https://netapp.audubon.org/cbcobservation/), detailing the information for Bogota, Cali and Medellin in the search parameters. Data can also be obtain from each of the local count coordinators at: Asociación Bogotana de Ornitología ABO: https://www.avesbogota.org/ abo@avesbogota.org Asociación para el estudio y conservación de aves acuáticas en Colombia CALIDIRS https://calidris.org.co/ calidris@calidris.org.co Sociedad Antioqueña de Ornitología SAO: https://sao.org.co/ sao@une.net.co.

**Funding:** J.D.A.E. received Funding was provided by Vicerrectoría de Investigaciones de la Pontificia Universidad Javeriana, Bogotá, no. 20101. The funders had no role in study design, data collection and analysis, decision to publish, or preparation of the manuscript.

**Competing interests:** The authors have declared that no competing interests exist.

habitat loss, and fragmentation [4]. Consequently, urban growth has affected the diversity and distribution of communities and populations of several biological groups. Urbanization usually involves a significant reduction in biological richness due to the disappearance of numerous native species that are less tolerant to novel urban conditions. However, urban growth also frequently promotes a higher abundance of a reduced group of species ecologically associated with humans (synanthropic species) that benefit from urban infrastructure and human activity [5, 6].

Bird species are some of the organisms most affected by urban growth [7]. Many studies have shown that urbanization affects avian community richness and abundance, as well as patterns of habitat use, foraging, and breeding [8]. Urbanization usually leads to a gradual loss of native bird species, while those considered synanthropic tend to be abundant and dominant. Regularly, these negatively impact habitat-specialist species associated with frugivorous and insectivorous diets as well as ground and shrub nesters [9]. Conversely, urbanization favors generalist species with granivorous and omnivorous diets and those capable of nesting in buildings [10].

Knowledge gaps persist despite many studies of bird communities and populations in urban areas [11]. For example, bird research in cities of the Neotropics in Latin America is still incipient and just beginning to generate information [12]. Although it is possible to find studies conducted on urban birds in Neotropical cities from the 1970s [13], more than 85% of such publications have been published in the last 15 years. Most of these publications focus on urban species lists and descriptions of the spatial distribution of some bird species in specific urban green zones like urban parks, lakes, wetlands, and remnants of native vegetation [11].

A significant gap in understanding bird patterns in urban areas in the Neotropics is related to a lack of long-term studies (e.g., [14, 15]). Most of the information about birds in cities from this region comes from studies that usually do not exceed 1–2 years [11]. Although these short-term studies show an initial overview of bird community diversity and spatial distribution in some Neotropical Latin American cities, they could be skewed toward more conspicuous and abundant species since they have a higher probability of detection. Such studies could limit the reports of rare, inconspicuous species, or those with specific patterns of occurrence, only recorded at particular times (e.g., bird migrant species with seasonal or altitudinal movements).

One of the few reliable multitemporal datasets of urban birds in Latin American cities corresponds to the Christmas Bird Counts (CBC) [16]. This program, led by a global effort of the Audubon Society, has endured for over 100 years, with more than 60,000 observers participating annually [17]. The CBC consists of bird counts for a maximal period of 24 hours, gathered between the second week of December and the first week of January in different sample sites located inside a circular area of 24 km diameter. Counts have been made regularly in Latin American countries, including various natural and anthropic habitats that are easily accessible (forests, wetlands, bamboo areas, urban and suburban areas, etc.). Thus, the CBC has been recognized as a valuable source of data for the region, providing information on biogeographical patterns, changes in spatial distribution, population ecology, and diversity of birds and their relationship to human activities [14, 15, 18].

Colombia, as the Neotropical country with the highest avian species richness in the world, has adopted this initiative and methodology with yearly counts since 1991 through the Asociación Bogotana de Ornitología (ABO). Since 2001 the CBC has been carried out regularly in more than ten places as part of the National Bird Censuses, including urban-rural settings in cities such as Medellín and Cali. Stiles et al. [15] used such information to evaluate temporal changes in the abundance of some species in Bogotá. However, no study has used this information to better understand the patterns of diversity and spatial distribution of urban avifauna at

the community level based on accumulated data in Bogotá and other cities of Colombia, where these counts have usually been carried out.

Using CBC multitemporal count data available from 2001–2018, we identified and analyzed bird accumulated richness, similarity, and complementarity patterns in three CBC circles located in and around three main cities of Colombia (Bogotá, Cali, and Medellin). We analyzed the patterns for the entire community for trophic and tolerance levels to urbanization guilds. The information is expected to contribute to understanding the diversity and distribution of urban birds in Neotropical cities and promote future studies on the relationship that urban variables at local and landscape scales have to these patterns. This will provide a baseline for decision-making and for recommendations allowing the reconciliation to the growing demand for urban infrastructure with the need to preserve native birdlife in megadiverse Neotropical countries like Colombia.

## Methods

### Study area

We used data available from the Asociación Bogotana de Ornitología—ABO, Sociedad Antioqueña de Ornitología—SAO and Asociación Calidris-Río Cali gathered in the last 18 years during the Christmas Bird Counts (CBC) (2001–2018) developed in and around the three largest cities in Colombia: Bogotá (Sabana de Bogotá Christmas Bird Count Circle—ABO), Medellín (North—SAO1 and South- SAO2 Christmas Counts Bird Circles), and Cali (Cordillera Occidental Christmas Bird Count—CCO) (Fig 1).

The ABO circle covers urban and rural areas in the northwestern sector of Bogotá (center: 4˚48'00.00" N, 73˚10'59.91" W), a city located in the Eastern Cordillera of Colombia at 2650 m [19] and inhabited by around 8 million people in an urban area of approximately 380 km$^2$. Bogotá has an average temperature of between 12˚C and 16˚C and an annual rainfall of 824 mm [19]. As an urban center, Bogotá has a long history of urbanization and profound changes in land use [20].

The SAO1 and SAO2 circles were merged in this study to enhance temporal and site coverage. They include an urban area of Medellín and the neighboring municipalities of Sabaneta, Itagüí, Envigado Angelópolis, Amagá, Caldas, La Estrella, Copacabana, Girardota, Medellín and Bello that make up the so-called Valle de Aburrá (center: 6˚22'00.00" N and 76˚24'00.00" W). Both circles are located at 1500m, in urban and rural areas with ca. 3,800,000 inhabitants [21] and an average annual rainfall of 1685 mm and average temperature of 21.5˚C [22, 23]. The Valle de Aburrá is the main industrial and manufacturing area of Colombia [24].

Finally, the CCO circle includes peri-urban and rural areas of Cali and neighboring municipalities of La Cumbre and Yumbo (center: 3˚56'66.53" N y -76˚57'63.48" W). The area borders the Farallones de Cali National Natural Park to the north and covers two important areas for the conservation of birds (AICA/IBA): "El Bosque de Niebla de San Antonio/Km 18 and the Zonas de Chicoral and Alto Dapa". Cali is Colombia's third most populated city, with 2 million inhabitants, an average temperature of 23˚C, and an annual rainfall of 900 mm [22]. It corresponds to southwest Colombia's largest and most economically important urban center and is considered one of the most ecologically diverse cities, given that its urbanization model seeks to maintain existing vegetation, expand green areas, and increase arborization [25].

The CBC followed the standardized methodology used by the National Audubon Society [17], in which the presence and abundance (bird registers reported) of birds are evaluated at different sample sites within the area comprising a circle of 24 km in diameter, during a maximum period of 24 hours and usually between December 14 and January 5. At each site, the birds seen and heard are recorded, as well as information on the observation effort (number of

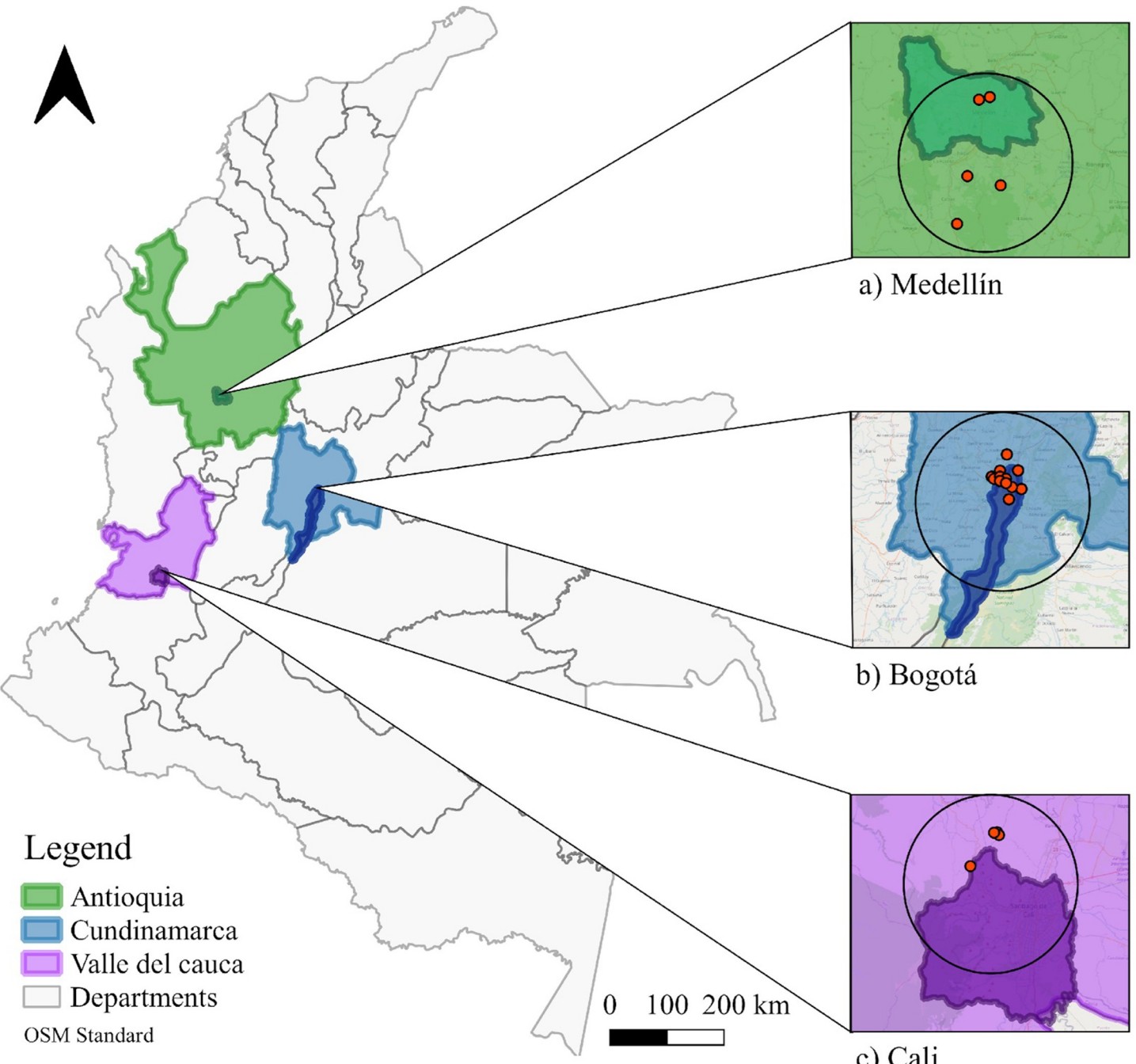

**Fig 1. Study area for the Christmas Bird Count (CBC) circles.** (a) Medellín (Antioquia) (green background), (b) Bogotá (Cundinamarca) (blue background) and (c) Cali (Valle del Cauca) (purple background). Orange dots represent sampling sites per city. Background images from https://sites.google.com/site/seriescol/shapes.

observers, distance traveled, duration of each count, and method of transport (S1 Table in S1 File) and weather conditions (cloudiness, temperature, and precipitation).

Data was first acquired from the Audubon Society's online database (c.f. https://netapp.audubon.org/cbcobservation/). Detail information (yearly and per site) was collected from each local association that coordinates each CBC circle (Asociación Bogotana de Ornitología -ABO, Sociedad Antioqueña de Ornitología–SAO).

## Bird data collection

The list of species recorded year to year in each sample site for each city circle was compiled, reviewed, and refined to detect potential typographical mistakes or misidentifications, as well as synonyms or taxonomic updates. Taxonomy and nomenclature were adjusted to the South American Classification Committee of the American Ornithological Society [26]. From the complete species list, the global conservation status was determined according to the Red List of Threatened Species [27] and the national risk based on the Red Data Book of Birds of Colombia Volume I and II [28, 29]. Species were also classified according to their migrant status (boreal or southern) or resident status [30]. Species accumulation curves were constructed using the EstimateS program, version 9.1.0 [31], to evaluate the representativeness of each community in each city circle and sample site. Using the non-parametric Chao 2 and bootstrap estimators, with 95% confidence intervals obtained from 1,000 random iterations, we estimated the expected richness and compared it to the observed richness.

To characterize the bird communities found between sample sites and city circles, we estimated patterns of richness (based on the number of species detected) and counts (based on the number of registers by each species detected). The estimations were made at the community level, encompassing all registers from all species recorded. We also calculated richness and individual count patterns according to two independent criteria. First, species were classified in guilds according to their response to urbanization, based on Fischer et al. [32] as: "avoiders," "utilizers," and "resistant" (dwellers), based on expert criteria and life history information (c.f. https://birdsoftheworld.org/bow/home). Second, species were classified in guilds according to their main food item, based on "Birds of the World" [33] (https://birdsoftheworld.org/bow/home), local field identification guides, and expert judgment. Categories for guilds were as follow: carnivores, frugivores, granivores, insectivores, nectarivores, and omnivores. Although classifying one or another species as a consumer of a single food item leaves out essential singularities [34], it offers an initial insight into the requirements and ecological variations of the species.

Some studies have used the number of participants per hour or party/hour to standardize the sampling effort. However, Sauer et al. [35] and others call for caution when using this, especially when not dealing with species-specific cases, incomplete effort information such as the total number of hours, distances, or observers [18, 35]. Since our data did underline some of these issues, we presented the data accumulated by participants per hour (number of individuals times the number of hours and observers in each city) as a comparison of the sampling effort of other counts [36], with no species-specific corrections. Also, information on the accumulated sampling effort in terms of distance traveled and time invested in the observation periods analyzed was presented (S1 Table in S1 File).

To quantify and compare the cumulative species richness of the bird community found in each city and their sample sites, individual-based rarefaction analysis was used, using the package iNext [37] in RStudio. Also, we evaluated and compared annual mean richness and individual count found within each site per city using a generalized linear model (GLM) with the glm function and the lme4 package [38]. The GLM was considered a Poisson error distribution for richness and a negative binomial error distribution for counts according to the over-scattering detected in the data with Logit as the identity link function. These analyses were done for the richness and counts of each site, the whole community, the categories of group responses to urbanization, and the main food categories.

From a non-metric multidimensional scaling (NMDS), similarity in the bird community composition between sites was determined using the total relative individual count of each species per year (for the whole community, the urban-response categories, and dietary

categories). Before running the NMDS, the three birds classified as invasive and escaped were excluded for Bogotá, aiming to have the responses of native species not exclusively associated with urban environments. The ordering was evaluated using the coefficient of determination based on stress $R^2$ [39], the goodness of fit with a squared correlation coefficient ($R^2$), and its adjustment through 999 permutations. The NMDS was performed using the metaMDS function in the vegan package [40, 41] in RStudio. With the NMDS similarity matrix we performed a similarity or similarity analysis (ANOSIM) and a PERMANOVA to test for significant differences in categories by sites in the PRIMER 7 software [42].

Finally, we evaluated the complementarity between bird communities at the different sites of each city. We relied on a nesting analysis between sites, using the "nestedness metric based on overlap and decreasing filling" (NODF) [43]. Nesting was calculated using the nestednodf function of the RStudio vegan package [40]. NODF scores ranged from 0–100, showing non-nested communities with higher complementary to those with perfect nesting and low complementary between sites.

## Results

For the 18 years of data in the three cities included in this study, we identified a total of 464 species for 105,302 individuals belonging to 54 families. The families with the highest number of species were Thraupidae and Tyrannidae with 62 species each, while the most recorded species were *Zenaida auriculata* with 13,255 individuals (12.6% of the individuals observed) and *Zonotrichia capensis* with 7,253 individuals (6.9% of individuals surveyed) (S2 Table in S1 File).

In the case of Bogotá, 83,710 individuals of 214 species from 44 families were recorded, and Thraupidae with 30 species and Tyrannidae with 27 species were the most representative. The most reports species was *Zenaida auriculata* (12,754 individuals, 15% of the total number of individuals observed), followed by *Zonotrichia capensis* and *Orochelidon murina*. The number of species varied from year to year, between 110 and 141 with the number of individuals between 3,523 and 64,74 (S2 Table in S1 File). For Medellín, during 14 years of counting, 10,142 individuals of 270 species were recorded (42 families). *Streptoprocne zonaris* was the most common species (510 individuals, 5% of all individuals), followed by *Zenaida auriculata* and *Pygochelidon cyanoleuca*. The most representative families were Thraupidae with 59 and Tyrannidae with 50 species. The number of species also varied yearly between 57 and 166 species, and the individuals in the circle between 1 and 510. For Cali, data was collected for 16 years with 11,490 individuals of 258 species (40 families). The most representative families were Thraupidae with 40 species, Tyrannidae with 36, and Trochilidae with 31 species. As in Medellín, the most counted species was *Streptoprocne zonaris* (1,090 individuals, 9% of the total individuals). Finally, the species recorded varied between 79 and 141 species, with 313 to 1,318 reports (S2 Table in S1 File). It is worth mentioning that 20 species have only been recorded once in Bogotá during the study, while 14 species account for two year-records. As for Medellín, 39 species hold one count in the 18 years and 20 two records. Cali has 47 with one record and 16 with two records. In all cases, these singletons or doubletons in each city represent a mix of species from residents, vagrants, and/or migratory species worth looking into in more detail in future counts and sites.

We identified 17 species in the three cities that were in some category of risk of extinction, either internationally or nationally. Critically Endangered (CR) *Cistothorus apolinari* and *Pseudocolopteryx acutipennis* were found and Endangered (EN) species such as *Cistothorus apolinari*, *Rallus semiplumbeus*, *Porphyriops melanops*, *Scytalopus stilesi* and *Oxyura jamaicensis*, and as Vulnerable (VU) *Chloropipo flavicapilla*, *Hypopyrrhus pyrohypogaster*, *Grallaria rufocinerea*, *Sericossypha albocristata*, *Chlorochrysa nitidissima*, *Creurgops verticalis*, *Ampelion*

*rufaxilla*, *Setophaga cerulea*, *Rallus semiplumbeus*, *Icterus icterus*, and *Tinamus tao* (S2 Table in S1 File). Bogotá registered the highest number of species classified in different IUCN conservation categories (7 species), followed by Medellín (5 species) and Cali (5 species) (S2 Table in S1 File).

The species, according to their residence status [30], show that for all the cities the resident species are predominant (S2 Table in S1 File). In Medellín and Cali, migratory species constitute 8.9% of species registered in the Christmas Bird Counts (e.g., *Pygochelidon cyanoleuca*; *Vireo leucophrys*), and in Bogotá, 18.2% (e.g., *Tringa flavipes*; *Spatula discors*). Bogotá is the only city where *Alopochen aegyptiaca*, an invasive species, was registered as well as two local migrant species, *Porphyrio martinica*, and *Heliornis fulica*, and two domestic species *Anser anser* and *Cairina moschata*, the latter shared with Medellín (S2 Table in S1 File).

### Representativeness and sampling effort

The representativeness of species estimated from the accumulation curves in each city showed a tendency for the asymptote with values of representativeness greater than 83% (Chao 2), as well as low probabilistic values (singleton and doubleton) (Fig 2 and S3 Table in S1 File).

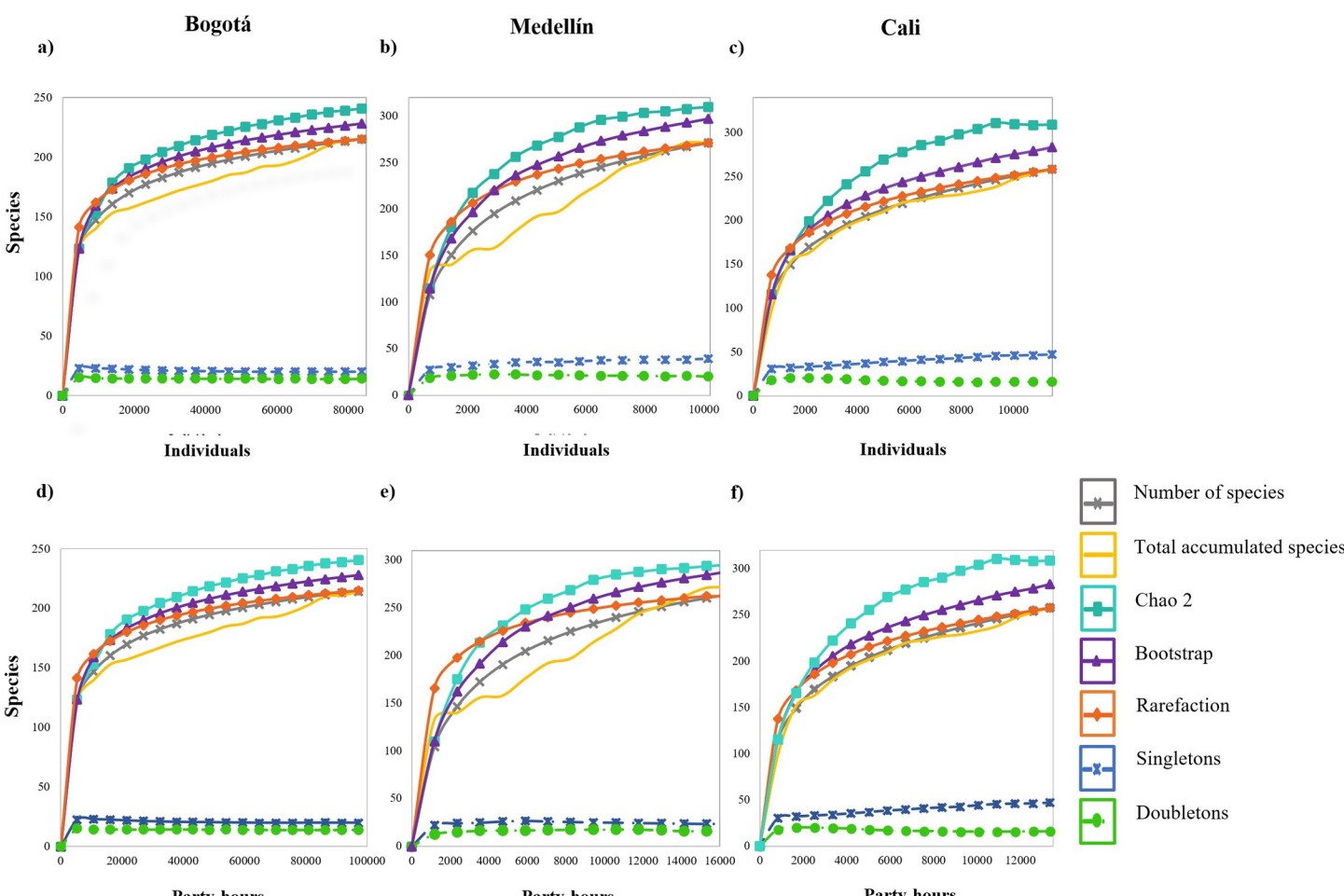

**Fig 2.** Upper panel: species/individual accumulation curves for each city between the years 2001 and 2018. (a and d) Bogotá (ABO). (b and e) Medellín (SAO). (c and f) Cali (CALIDRIS). Upper panel (a-c) species rarefaction curves, Lower panel: Party-hours for each city between 2001 and 2018. Number of species (yellow line), number of cumulative species (grey), estimated rarefaction (orange), Chao2 (green), Bootstrap (purple), Singletons (blue), Doubletons (lawngreen).

Bogotá showed the highest representativeness with 89.3% (Fig 2A), followed by Medellín at 87.7% and Cali with 83.5% (Fig 2B and 2C).

Despite the differences in the sampling effort between the cities, comparing them through party-hours, no differences were observed in the expected total species richness. Based on this sampling-effort comparison in all cities, the representativeness was above 87.7%, although the number of hours and participating observers varied widely (Fig 2D and 2E and S3 Table in S1 File).

### Richness and individual count

According to the rarefaction analysis, sites in Cali showed the highest estimated cumulative richness (mean: 221.4; sd: 20.64) compared to Medellín (mean: 158.61; sd: 34.09) and Bogotá (mean: 111.9; sd: 22.9) (Fig 3). In Cali, peri-urban sites with extensive forest cover, such as Chicoral-Dapa (CHIDA), maintain higher richness with 252 species, and the lowest, as is the case of Chicoral-Escuela (CHIES) with 190 species (Fig 3C). In Medellín, El Parque Ecológico Recreativo Alto de San Miguel (PERASM), located in the rural area of the city, showed the greatest richness with 193 species, while a green urban area, Jardín Botánico de Medellín (JB), registered the lowest richness with 111 species (Fig 3B). In Bogotá localities associated with urban wetland habitats showed the highest estimated richness. Such as Humedal Córdoba-Parque Niza (HCOR) with 136 species and Humedal La Conejera (HCON) with 122 species, while green areas in urban and peri-urban sites such as Valle de Teusaca (VTE) with 72 species and Parque Simón Bolívar (PSB) with 100 species were those that show the overall lower richness (Fig 3A) (S4 Table in S1 File).

The average annual richness showed a similar trend to those findings in the accumulated richness, where Cali is the city that annually registers a greater number of species. Within each town, differences are observed regarding which sites report the highest number of species annually, with some trend towards peri-urban areas. For example, in Cali, the average richness was higher in Km 18 (KM18) (79.7±2.8) (Fig 4C), in Medellín, in the Parque Ecológico Romera (PER) (67±2.9) (Fig 4B), and in Bogotá, in Aurora Alta (AUA) (51.5±1.7) and Tabio (TBO) (40.1±1.5) (Fig 4A).

Annual average bird count was different from the richness results. Bogotá was the city that showed the highest values, particularly at wetland sites such as Humedal Jaboque (HJB) (792.5 ±6.8) and with the lowest records in TBO (274.2±3.9) and AUA (274.2±3.9) (Fig 4A). In Cali, species abundance was similar between sites (Fig 4C). For Medellín, the sites with the highest reports corresponded to the same sites that presented the highest average annual richness (PER and PERASM) (Fig 4B).

For all cities, avoiders had a higher richness annually and overall. Cali and Medellin with higher avoider richness in KM18 and PER sites (49.5±2.2), and Bogotá at TBO (33.7±1.4) (S5 Table in S1 File). These three sites within each city can be considered more peri-urban (S1 Table in S1 File). For average bird counts, Bogotá differed from the other two cities in that dwellers showed a higher reports at various sites, contrary to avoiders as the most common (S5 Table in S1 File).

Insectivores showed the highest richness and mean counts across all cities. Cali showed the top insectivore richness in KM18 (33.4±1.8), followed by Medellín in PER (29.4±1.9) and Bogotá with HJB (424.6±5). By number of individuals, Bogotá had the highest mean abundance of insectivores in HJB (424.6±5), followed by Cali in Chicoral-Montebello (CHIMO) (118.1±2.9), and Medellín in PERASM (110.1±3.7) (Fig 4) (S5 Table in S1 File).

### Similarity and complementarity

The analysis of similarity of the bird communities recorded between 2001 and 2018 in the three cities confirmed quantitative differences found in richness and bird counts while

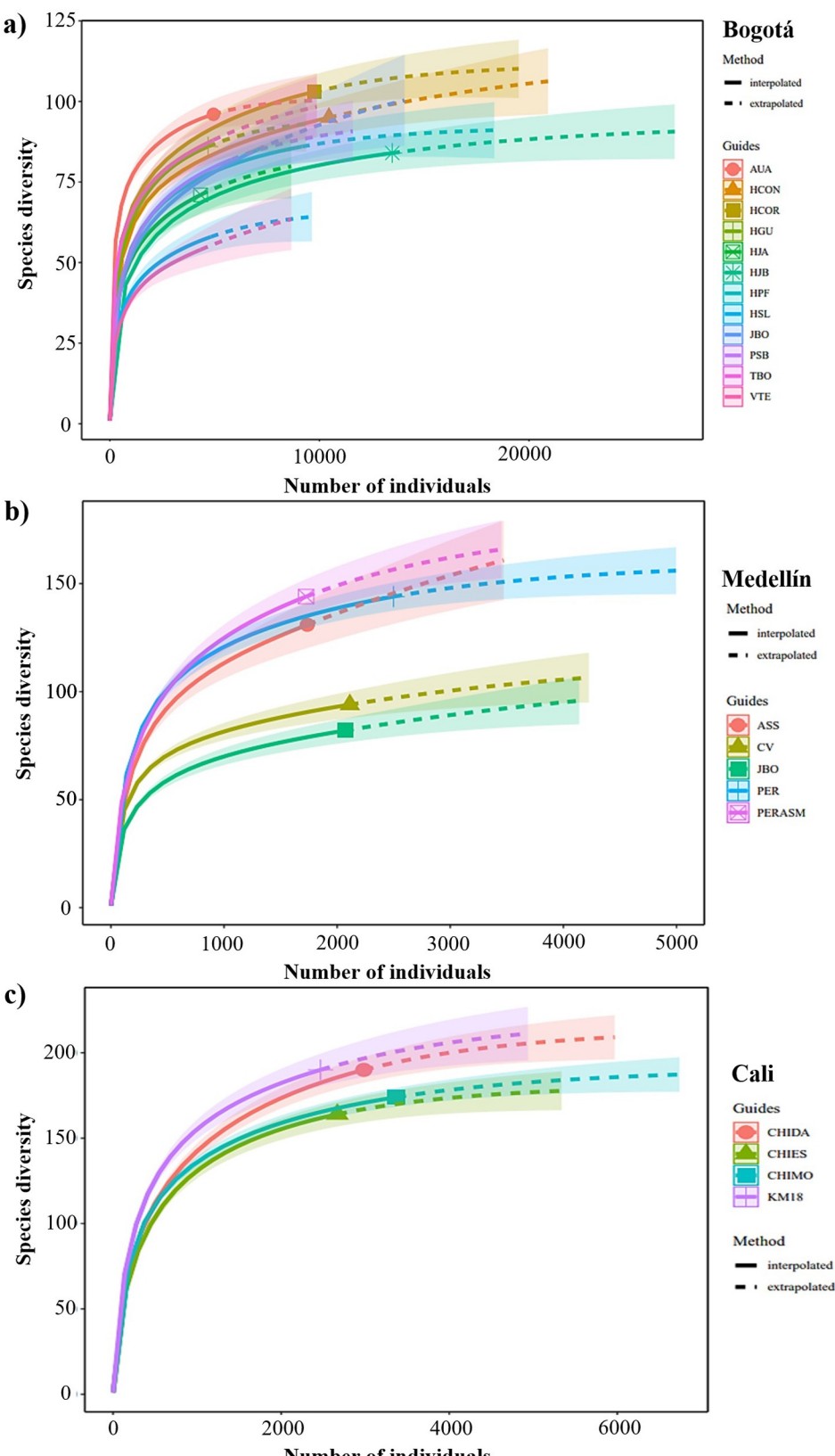

**Fig 3. Species rarefaction curves for each city with their respective sites between 2001 and 2018.** (a) Bogotá (ABO), (b) Medellín (SAO), and (c) Cali (CALIDRIS). Sampling sites in Bogotá: Aurora Alta (AUA), Humedal de Córdoba-Parque Niza (HCOR), Humedal de Guaymaral (HGU), Humedal Jaboque (HJB), Humedal Juan Amarillo (HJA), Humedal la Conejera (HCON), Jardín Botánico (JBO), Parque La Florida (HPF), Parque Simón Bolívar (PSB), Santa María del Lago (HSL), Tabio (TBO), Valle de Teusacá (VTE). Sampling sites in Medellín: Cerro el Volador (CV), Jardín Botánico de Medellín (JB), Alto de San Sebastián (ASSE), Parque Ecológico La Romera (PER), Parque Ecológico Recreativo Alto de San Miguel (PERASM). Sampling sites in Cali: Chicoral-Dapa (CHIDA), Chicoral-Escuela (CHIES), Chicoral-Montebello (CHIMO), Kilómetro 18 (KM18) (Details in S1 Table in S1 File).

showing qualitative differences in the species, both at a general level and concerning the urbanization response (Fig 5) and diet guilds (S1-S3 Figs in S1 File). These qualitative differences were evident in Bogotá and Medellín.

In Bogotá, the sites located in peri-urban areas of the city (AUA and TBO) showed a high degree of dissimilarity to the remaining ten sites in the city (Fig 5A), mainly related to species considered avoiders (stress: 0.95), and those with frugivorous diets (stress: 0.95) (e.g., *Turdus ignobilis; Stilpnia vitriolina*) (S1 Fig in S1 File). In Medellín, dissimilarities were observed between those sites in peri-urban areas (ASSE, PER, and PERASM) related to avoider species (Fig 5B) (e.g., *Turdus ignobilis; Stilpnia vitriolina)* (stress: 0.99), but with insectivorous diets (e.g., *Henicorhina leucophrys; Myioborus miniatus*) (stress: 0.99) (S2 Fig in S1 File). In contrast, in both cities high similarity in bird communities was observed at all urban sites, mainly due to the presence of common dweller (stress: 0.99) and utilizer (stress: 0.98) species with granivorous diets (e.g., *Zenaida auriculata; Amazona ochrocephala*) (Fig 5B and S2 Fig in S1 File).

In the case of Cali, we found high similarity at all the sites (Fig 5C) (stress: 0.93) on the level of urbanization tolerance guilds (stress: avoiders: 0.94; dwellers: 0.94; utilizer: 0.94) and eating habits (stress: carnivores: 0.99; frugivores: 0.94; granivores: 0.95; insectivores: 0.93; nectarivores: 0.95; omnivores: 0.99) (S3 Fig in S1 File).

Complementarity (measured by the degree of nesting) in the composition varied at the city level and according to the response guilds to urbanization and diet (Fig 6). Medellín was the city that showed a greater general complementarity between sites (46.56) (Fig 6B), while Bogotá (65.41) (Fig 6A) and Cali (62.03) (Fig 6C) showed a lower complementarity that was similar between them. The species considered as avoiders with frugivorous diets contributed to a greater extent to the complementarity of the bird communities of each city, particularly in Bogotá (avoiders: 39.62, frugivorous: 54.17) and Medellín (avoiders: 43.79, frugivorous: 41.44), although in the latter city, the omnivorous species also showed the greatest complementarity of all the dietary guilds. In contrast, species considered dwellers and carnivores or granivores showed the least complementarity, especially in Bogotá (dwellers: 64.97, granivores: 76.8) and Cali (dwellers: 76.11, granivores: 56.84) (Fig 6A–6C).

## Discussion

The analysis presented here showed how long-term CBC data can be analyzed over time to recognize with more precision the composition and structure of bird communities within and around urban areas [8]. Avian monitoring involves systematically collecting and analyzing long-term data under a question or premise of analysis that consolidates complete inventories over time and recognizes their changes and/or patterns on a spatial and temporal scale, as well as using the results to guide the formulation and evaluation of conservation policies [44]. Thus, CBC long-term community characterization made it possible to identify patterns of attributes such as richness, abundance, similarity, or complementarity in bird communities and how these might vary at a spatial level, making it possible to obtain an increasingly

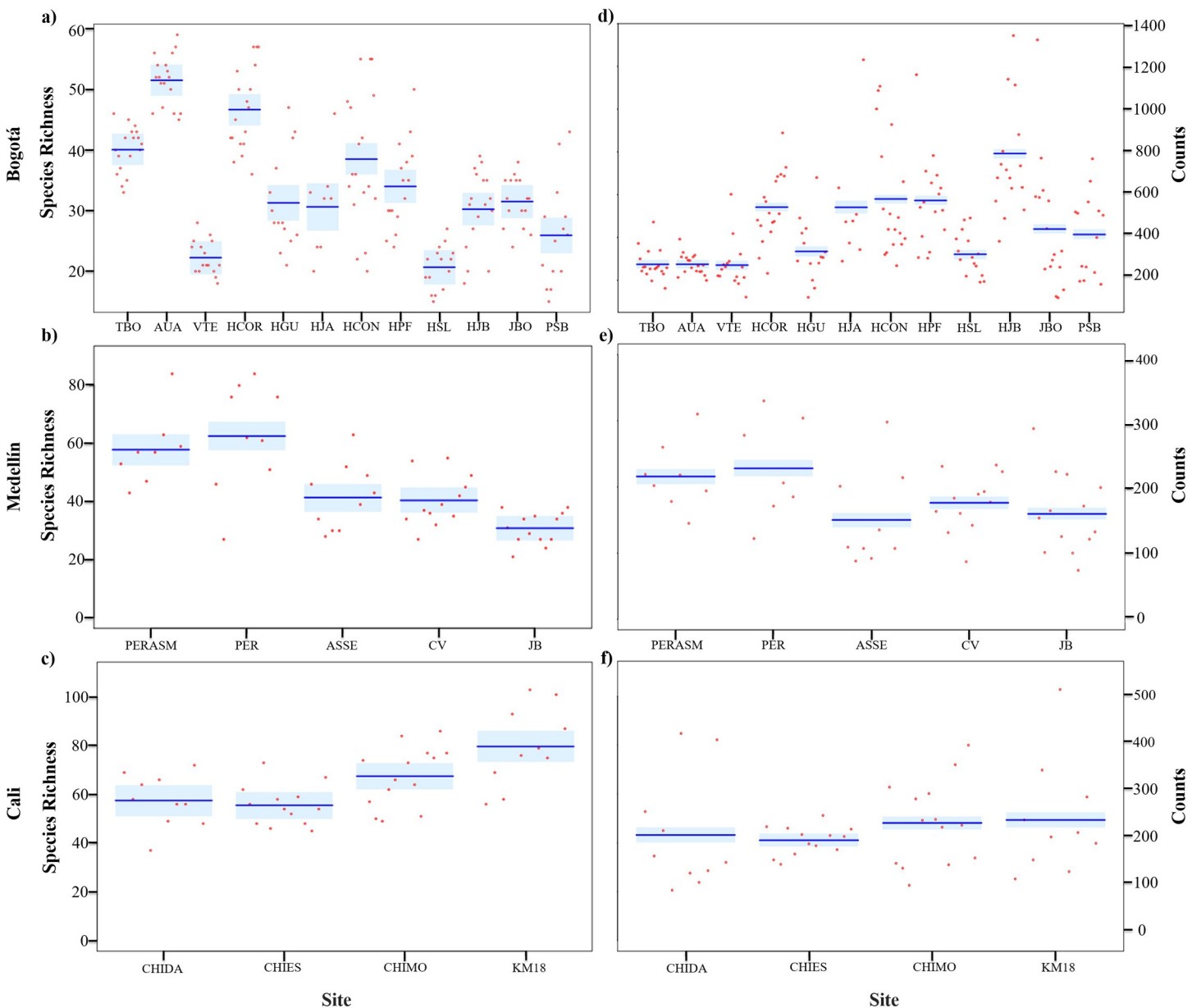

**Fig 4. Estimated richness and counts of birds by site in the three cities evaluated during the 18 years of the Christmas Count in Colombia.** Bogotá (a and d), Medellín (b and e), Cali (c and f). Left panels: species richness, right panels: individual count. Variation in annual estimates by location is represented by red dots. The dark blue line represents the mean of the estimate produced by the GLM, and the light blue areas are the 95% confidence intervals of the estimate. Sampling locations in Bogotá: Aurora Alta (AUA), Humedal de Córdoba-Parque Niza (HCOR), Humedal de Guaymaral (HGU), Humedal Jaboque (HJB), Humedal Juan Amarillo (HJA), Humedal la Conejera (HCON), Jardín Botánico (JBO), Parque La Florida (HPF), Parque Simón Bolívar (PSB), Santa María del Lago (HSL), Tabio (TBO), Valle de Teusacá (VTE). Sampling locations in Medellín: Cerro el Volador (CV), Jardín Botánico de Medellín (JB), Alto de San Sebastián (ASSE), Parque Ecológico La Romera (PER), Parque Ecológico Recreativo Alto de San Miguel (PERASM). Sampling locations in Cali: Chicoral-Dapa (CHIDA), Chicoral-Escuela (CHIES), Chicoral-Montebello (CHIMO), Kilómetro 18 (KM18).

complete picture of the diversity and distribution of bird communities that persists in urban and peri-urban environments in cities of the Neotropics [18, 45].

In Colombia, the CBC has been done in 93 circles since the year 2000. Colombia is the country with the most circles outside the United States and Canada and confirms the importance of these participatory bird monitoring processes as a source of information [46].

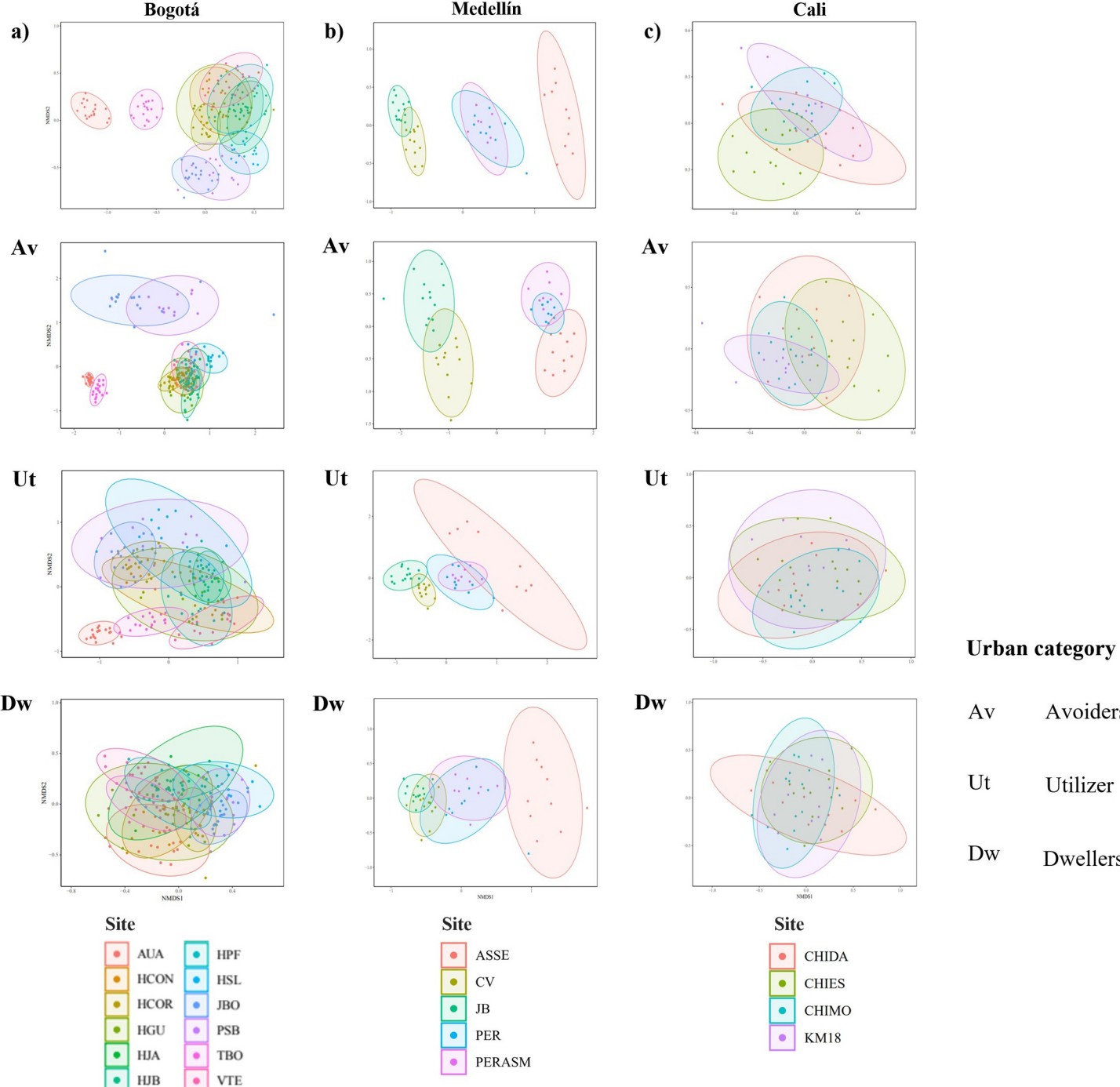

**Fig 5. Non-metric multidimensional scaling analysis (NMDS) based on individual count of the species reported between 2001 and 2018 in the three Christmas Count cities in Colombia.** Av: avoiders, Ut: utilizers, Dw: dweller (sensu Fischer et al. [33]). All site nomenclature as in previous figures and S1 Table in S1 File.

However, Colombia also exemplifies the limited use that has been made of this information at the Neotropical scale. Research projects such as those of Stiles et al. [47], Stiles et al. [15], and Quiñones et al. [48] are some of the few that have taken advantage of this comprehensive data

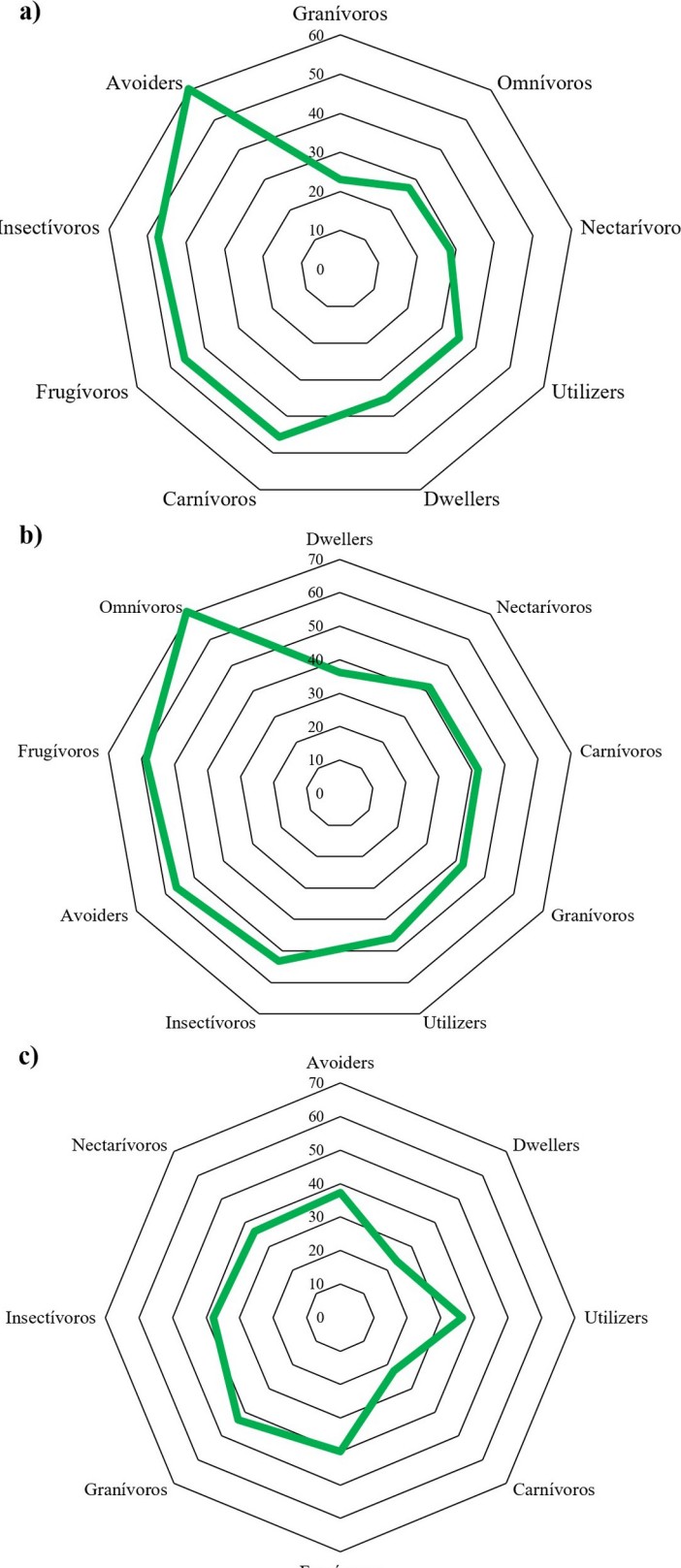

**Fig 6. Nestedness analysis (NDOF) that indicates the complementarity at the city and guild scale of response to urbanization and diet in the composition of the species recorded in the count sites evaluated over 18 years of Christmas Count.** (a) Bogotá, (b) Medellín, and (c) Cali.

source with great potential for understanding the responses of the avifauna to the growth of cities as well as for guiding conservation processes at this level [45].

Based on CBC data, we have been able to characterize the avian communities in the three main cities of Colombia, despite having different histories of development, ecosystems, and multiple biophysical variations. In doing so, we recognize as permanent or temporary almost a quarter of the species registered in Colombia (464 of 1954 species) [49]. Similarly, we have assessed differences in the urban avian communities based on their possible tolerance to urbanization and with a wide range of diets that persist in these anthropized spaces, indicating high levels of complementarity between sites and cities far from showing a homogenized avifauna in Neotropical cities.

Although comparing the three cities is not feasible, nor does it correspond to the scope of this study, we found similar trends relevant to the diversity and distribution of the avifauna in them. First, the representativeness of the community obtained through the counts carried out for 18 years exceeds, in all cases, 84%, showing values of species richness that range between 214 and 278 species in the three cities. These values also exceed the number of species that have been identified in counts year after year (108 species on average), as well as the representativeness and richness found in short-term studies in these same cities [50, 51]. The latter is evident when analyzing the richness observed and estimated in this study with previous characterizations of the avifauna developed in the same cities. For example, in Bogotá, 214 species have been reported in the CBC (Chao: 240 species) in 18 years and at 12 count sites, 91% (235) of those identified by Stiles et al. [15] and 94% (234 species) of those reported in Stiles et al. [47], both studies carried out in 26 years of counting and 29 study sites. In Medellín, with 270 species with the CBC (Chao: 309 species), we identified double the species reported by Garizábal-Carmona and Mancera-Rodríguez et al. [50] (83 species) in one year and 44 sampling points, but about half of those in Estrada [52] (445 species) based on long-term data from SAO observation lists, museum records, inventories and information from the GBIF (Global Biodiversity Information Facility). Finally, in Cali, the CBC data reports an intermediate richness of 258 species, 46% (Chao: 308 species) of the 561 species recorded by Palacio et al. [53] from literature reviews, scientific collections, and eBird. Despite some differences relating more to using complementary sources of information, these results confirm the value of systematic monitoring such as those promoted by the CBC.

The value of the information that the CBC can provide in the long term is also related to a continuous tracking of the avifauna in areas considered to have a high ecological value in cities or for detecting the presence of groups of species with some conservation concern or interest [54]. The three cities included in this study have areas identified as ecological important for bird diversity conservation, such as "El Bosque de San Antonio/Km18", in Cali, a Biodiversity-IBA (one of the few sites where ornithological studies have been carried out by over 100 years) [55, 56]. In Medellín, Cerro "El Volador" is a conservation priority area and one of the few urban areas with high species richness [57]; while, for Bogotá, the relics of wetlands stand out, currently in the RAMSAR category [58]. These areas protect the 17 species with different categories of extinction risk, according to the IUCN and the "Red Book of Birds of Colombia", like the aquatic species within the wetlands of Bogotá, where the greatest number of species at risk was found [59], and "La Romera" in Medellin that could act as a refuge for threatened species (A. Castaño pers. comm.). Similarly, migratory species also use urban settings to some extent in the three cities. Based on the CBC data, we confirm 47 migratory bird species in the three cities.

The number of bird species observed yearly showed wide variation in the sample sites. We found that urban habitats had more dwellers of insectivorous and granivorous species and individuals. Frugivores were relegated to fewer sites and coincided with avoiders, similar to previous studies [7, 60]. Hand in hand with this, we found that Bogotá had high values of complementarity between sites due to the concentration of avoiders in peripheral sites, similar to what we saw in Medellín. Conversely, the high similarity found in Cali among all its sites in all bird groups according to their tolerance to urbanization is possibly related to all sites being more peri-urban or rural. These results highlight the importance of the mosaic of areas as refuges for species with conservation concerns. They also highlighted the urgency for actions to prevent their degradation since they offered habitat for maintaining this avifauna [61]. However, differences within each city might also be related to factors that were not analyzed in this study, including the spatial configuration of the green areas found in these sample sites, such as their size, shape, or their connectivity, or aspects related to the quantity and quality of the habitat they offer [62]. Likewise, factors that characterize the urban matrix in which bird count sites are immersed (e.g., building density or urban noise) could influence limitations in their ability to retain some species [63]. In future studies, such spatial or temporal information should be included as urban covariates to enhance analysis capabilities (i.e., GLMs, GLMMs, or hierarchical models) that allow the identification of the size of this covariates effect and its contribution to explaining some of the bird community patterns found, as already described in other short-term studies in neotropical cities [64–66].

Some research has shown that larger cities with a higher degree of urbanization lose highly specialized groups and maintain species that are ecologically similar to each other, a process known as biotic homogenization [66, 67] that would not be supported in Bogotá and Medellín, accompanied by the high levels of complementarity between the sites in each city. Bogotá, the city with the most extensive urban area and fewer peri-urban sites has the lowest species richness but the largest number of green spaces and wetlands [19]. At the same time, other studies indicate that in an urban-rural gradient, the richness and diversity of birds reaches their maximum point at intermediate levels of urbanization [68] that could contribute to understanding why Cali has such high numbers per site, given that the sampling sites are located entirely in peri-urban or rural areas.

Spatial trends in this study reflect the differential responses of avifauna in urban environments in the Neotropics, where the premise of planning complementary spaces that contribute not only to species considered avoiders but also to avifauna composition is highlighted [69]. In Bogotá, widespread and previously common species such as *Spinus spinescens*, considered a dweller-granivore, are declining and others, apparently less common, such as *Zonotrichia capensis*, are increasing [15]. While conservation efforts prioritize rare and threatened species, marked declines in highly recorded species have significant implications for ecosystem functioning [70, 71]. We recognized that the bird community in each city had been temporally variable. This may present different dynamics in response to biophysical changes, such as those proposed by Stiles et al. [15] concerning climate change, mainly mean, the minimal and maximal temperature at one climatic station, with the strongest increasing tendency in the minimal annual mean temperature. But these could also respond synergistically to the anthropic dynamics that favor some species (dwellers) while isolating others (avoiders) [72, 73]. Since Stiles et al. [15] did not evaluate in any detail cover changes within Bogotá, the lack of clear drivers for avifauna change highlights the need to continue monitoring bird diversity in the long term in Neotropical cities, to increase its representativeness and achieve a better description and understanding of changes at the community level [74]. To this extent, the work carried out by the different associations and volunteers participating in the CBC in several urban and peri-urban areas in the Neotropical region offers a study window for multiplying

questions of urban ecology. Specifically, concerning the avian response to urban transformations in the world region where currently more than 80% of the human population is concentrated in cities [3].

We are aware that the results shown are biased by imperfect sampling. Biases include non-random distribution of the circles and sites, variability of the counting effort within and between circles [18], and in some cases, the incompleteness of the data (such as missing years for some sites or no information on sampling effort for years or cities) for research. In Colombia, the selection of monitored areas and sites may be due to their accessibility, future forecasts of change, or particular interests (e.g., personal or group interests of the ABO, Calidris, SAO, and other ornithological associations), for which there may not be an underlying ecological basis. The data and sampling effort have not remained stable during the 18 years of CBC for any circle, with information gaps for entire years, groups of observers varying from year to year, and the lack of consistently associated metadata for many years (distances traveled, weather, hours of observation, number of observers, and observer skill). We consider this might be somehow corrected by the temporality of the sampling, which, as identified in the analysis of species accumulation curves, shows representativeness above 84% despite the heterogeneity of coverage even within each city.

However, the organizations and groups that coordinate the CBCs in countries such as Colombia must strengthen rigorous data and metadata collection associated with sampling conditions. This information can help to better understand, for example, individual species fluctuations by relating them to biotic and abiotic variables, as well as species detectability, among others [75].

The results of this study showed the relevant role that citizen monitoring, such as those promoted by the CBC, can have in developing a complete analysis of diversity and distribution patterns of avifauna and other taxa. This information is critical for understanding spatial trends in bird community richness, frequency, or composition. Likewise, this study contributes to recognizing the significant variability that characterizes the aforementioned spatial trends. Such variations and their origins need to be addressed by understanding the factors and mechanisms that may be promoting them at the local and landscape scale. Spatial and temporal trends in urban avifauna will be critical to guiding planning processes in which it is possible to reconcile urban development and biodiversity conservation. This article seeks to contribute to this in the context of cities located in a megadiverse region such as the neotropics, where this discussion may become more urgent.

## Supporting information

**S1 File. Supporting files (Tables (1–5) and Figs (1–3)) including species lists per sites, metadata available, accumulation curves for study area, rarefaction curves, and non-metric multidimensional scaling analysis (NMDS) per city.**
(DOCX)

## Acknowledgments

We thank the Asociación Bogotana de Ornitología -ABO-, the Asociación Calidris, the Sociedad Antioqueña de Ornitología -SAO-, and the Sociedad Caldense de Ornitología -SCO-national coordinators of the censuses evaluated here, especially we thank Rocio Espinosa, as well as to the Audubon Society, in particular to Gloria Lentijo, for access to her individual and collective records of the CBC. We like to acknowledge each and every one of the volunteers who have participated in the censuses during these 18 years.

## Author Contributions

**Conceptualization:** María Angela Echeverry-Galvis, Juan David Amaya-Espinel.

**Data curation:** María Angela Echeverry-Galvis, Pabla Lozano Ramírez.

**Formal analysis:** Pabla Lozano Ramírez, Juan David Amaya-Espinel.

**Funding acquisition:** Juan David Amaya-Espinel.

**Investigation:** María Angela Echeverry-Galvis, Juan David Amaya-Espinel.

**Methodology:** María Angela Echeverry-Galvis, Pabla Lozano Ramírez.

**Project administration:** Juan David Amaya-Espinel.

**Supervision:** Pabla Lozano Ramírez, Juan David Amaya-Espinel.

**Writing – original draft:** María Angela Echeverry-Galvis, Pabla Lozano Ramírez, Juan David Amaya-Espinel.

**Writing – review & editing:** María Angela Echeverry-Galvis, Pabla Lozano Ramírez, Juan David Amaya-Espinel.

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
