## [Decision Letter · Decision Letter 0]

14 Oct 2022

PONE-D-22-20926Long-term Christmas Bird Counts describe Neotropical urban bird diversityPLOS ONE

Dear Dr. Echeverry Galvis,

Thank you for submitting your manuscript to PLOS ONE. After careful consideration, we feel that it has merit but does not fully meet PLOS ONE’s publication criteria as it currently stands. Therefore, we invite you to submit a revised version of the manuscript that addresses the points raised during the review process.

We look forward to receiving your revised manuscript.

Kind regards,

Daniel de Paiva Silva, Ph.D.

Academic Editor

PLOS ONE

Journal Requirements:

“J.D. A.E received Funding was provided by Vicerrectoría de Investigaciones de la Pontificia Universidad Javeriana, Bogotá, no. 20101.”

“Funding was provided by Vicerrectoría de Investigaciones de la Pontificia Universidad Javeriana, Bogotá, no. 20101.

The authors have declared that no competing interests exist.”

5. We note that Figure 1 in your submission contain map images which may be copyrighted. All PLOS content is published under the Creative Commons Attribution License (CC BY 4.0), which means that the manuscript, images, and Supporting Information files will be freely available online, and any third party is permitted to access, download, copy, distribute, and use these materials in any way, even commercially, with proper attribution. For these reasons, we cannot publish previously copyrighted maps or satellite images created using proprietary data, such as Google software (Google Maps, Street View, and Earth). For more information, see our copyright guidelines: http://journals.plos.org/plosone/s/licenses-and-copyright.

Additional Editor Comments:

Dear Dr. Galvis,

After this first review round, I believe your manuscript may be accepted for publication in PLoS One after you perform the imrpovements raised by the reviewers. Specifically, I think reviewer #2 has an important point regarding the lack of a clear hypothesis. If this issue is taken care of, and the other issues are solved, I believe the study will be suitable for pubication. Please resubmit the new version of your manuscript by January 15th 2023. Do not hesitate to resubmit earlier in case you are able to. If you have any additional doubt, please do not hesitate to write me.

Sincerely,

Daniel Silva

Reviewers' comments:

Reviewer's Responses to Questions

**Comments to the Author**

1. Is the manuscript technically sound, and do the data support the conclusions?

Reviewer #1: Yes

Reviewer #2: Partly

2. Has the statistical analysis been performed appropriately and rigorously? 

Reviewer #1: Yes

Reviewer #2: Yes

3. Have the authors made all data underlying the findings in their manuscript fully available?

Reviewer #1: Yes

Reviewer #2: Yes

4. Is the manuscript presented in an intelligible fashion and written in standard English?

Reviewer #1: Yes

Reviewer #2: Yes

5. Review Comments to the Author

Reviewer #1: I would like to congratulate the authors for the effort to collect data and analyze, works like this are of fundamental importance and relevance. I have some suggestions for improvement and clarity of the text.

Table S1: Duration is in which unit?

F1: Put the font of the shapes used to make the map. What are the red points?

Methods

How was the data that was used acquired? This must be in your methodology, I was even curious because in line 554 you talk about the inaccessibility of the data. So how were they acquired?

It would be interesting to have a better description of how the data was collected, it was not clear if the samples are in fact independent, for example, can two observers be observing the species in a nearby area? In IPA sampling, the listening points are considered independent at a distance of approximately 250m from each other. Two people sampling close together could be falsifying the abundance of the species, how did you work this issue?

line 538: summarize this biophysical changes proposed by Stiles et al. (2017) concerning climate change, this is interesting information to be in the text and it was one of my questions when I saw your results

As we do not have a table with a list of the species that were found in each year, it would be interesting for you to comment more on this variation, for example, in the XX year the Y species was not detected, this species is migratory and this year it is possibly its period of displacement was affected by El Niño or La Niña, or simply because this year the number of observers was lower and possibly this species may not have been sighted or detected.

Reviewer #2: The authors use information from the Christmas Bird Count over 18 years for urban and rural areas of three cities in Colombia and compared the different communities using richness, abundance (counts), and trophic guilds. The paper is mostly descriptive and does not include complex community models. I found that the authors did not have hypothesis about the study system. The descriptive information is interesting, but it will be better to have hypothesis testing. My suggestion is that the authors think about why the differences between the regions occur and try to use GLMs/GLMMs to explain the differences using local covariates (the urbanization degree, for example). Also, the authors could explore the differences among years in the number of species recorded, and I highly recommend using models that include estimation of parameters corrected by imperfect detection (hierarchical community models, for example). The authors acknowledged the importance of considering the sampling biases in the discussion, but they did not include any analysis in the rest of the manuscript. Finally, the figures and tables need much improvement.

Detailed comments:

Figure 1: The color of the different departments is orange and it’s not possible to distinguish which is each one. The authors need to explain the meaning of the red dots in the maps. Is it the sampling locations?

Line 185: I would not say that the abundance is recorded. Abundance is an estimated parameter. The observers recorded the counts, not the abundance.

Figure 2: Needs improvement. The authors should add a legend in figure showing what each color mean, even if it’s in the figure description. Also, the sites’ names should be in the top of each panel to make the interpretation easier. The authors need to take into account that some readers will only take a look at the figures, so they need to be as clear as possible.

Figure 3: Needs more information. The acronyms of each site should be explained in the figure description. Add the name of the cities in the top of each panel.

Figure 4: Hard to read because the axis font is too small. Needs improvement.

The ‘Study area’ topic fits better if inside the overall ‘Methods’ topic.

Lines 361-366: Again, abundance is an estimated parameter. The authors are referring to counts, not abundance in this paragraph. It’s hard to compare raw counts from places with different environmental characteristics.

Table 1: Really hard to understand. The authors need to organize better this table. Does dividing the information by site really matters? What does the letters in the side of the numbers mean? Consider add this table in the supplemental material.

Line 445: Better not start a paragraph with ‘However’.

Figure 5: Add the name of the cities and the urban category description in the figure too. The acronyms make it harder to understand.

6. PLOS authors have the option to publish the peer review history of their article (what does this mean?). If published, this will include your full peer review and any attached files.

Reviewer #1: No

Reviewer #2: **Yes: **Viviane Zulian

---

## [Author Response · Author response to Decision Letter 0]

25 Nov 2022

REVIEWER #1

Table S1: Duration is in which unit?

We have included the corresponding metric in “hour:minutes:seconds” as the means to determine duration.

F1: Put the font of the shapes used to make the map, and what are the red points?

Source for background images has been added to the figure legend, and the information regarding the red dots within the map.

How was the data that was used acquired? This must be in your methodology

We have clarified the data origin for all cities in the first paragraph of the methods section at the end (after lines 190).

I was even curious because in line 554 you talk about the inaccessibility of the data.

We adjusted to clarify which data was inaccessible and, as such, consider some of the caveats of the analysis. Such clarifications are now after line 623.

It would be interesting to have a better description of how the data was collected, it was not clear if the samples are in fact independent. For example, can two observers be observing the species in a nearby area? In IPA sampling, the listening points are considered independent at a distance of approximately 250m from each other. Two people sampling close together could be falsifying the abundance of the species, how did you work this issue?

We have included information regarding the minimum distance between the closest sites within each circle in Supporting Table 1, showing the minimum distance between sites per circle. The closest distance between any two points in any circle is no lower than 803 meters. However, considering that no between-site comparisons are made per city in terms of counts but rather presence-absence, we do not foresee possible double counts as an issue.

Line 538: summarize this biophysical changes proposed by Stiles et al. (2017) concerning climate change, this is interesting information to be in the text and it was one of my questions when I saw your results.

We have included (along lines 638 and onward in the new version) a few comments on the key conclusions of the cited manuscript to give readers a perspective.

As we do not have a table with a list of the species that were found in each year, it would be interesting for you to comment more on this variation. for example, in the XX year the Y species was not detected, this species is migratory and this year it is possibly its period of displacement was affected by El Niño or La Niña or simply because this year the number of observers was lower and possibly this species may not have been sighted or detected.

Data on the list of species found per city can be found entirely in supporting Table 2, under the last three columns regarding the city of the report.

We feel that given the lack of complete metadata on count registers, noticing the absence of the presence of a particular species could be misleading.

Unfortunately, site coordinators did not record every year information for observers, for example. In such cases, stating that the change in individual species count is due to one or another variable remains to be put to the test through other statistical and analytical methods (as suggested by Reviewer #2). The same applies to external or abiotic factors, such as climatic parameters. Therefore we do not feel comfortable extrapolating our data to such conclusions.

We have included a mention of the total number of species with very few counts in each city to call attention to colleagues regarding possible reasons for such low or inconstant counts (along lines 311).

REVIEWER #2

The authors use information from the Christmas Bird Count over 18 years for urban and rural areas of three cities in Colombia and compared the different communities using richness, abundance (counts), and trophic guilds. The paper is mostly descriptive and does not include complex community models. I found that the authors did not have hypothesis about the study system. The descriptive information is interesting, but it will be better to have hypothesis testing.

Indeed such is the case, as was also noted by Reviewer #1. 

This manuscript does not aim to determine the possible divers or causes of avifauna change in a comparative frame for temporal (years) or spatial variables (between cities). 

We first proposed the analysis to identify how long-term monitoring studies are essential for better precision and representativeness of urban bird communities' spatial and temporal patterns. Additionally, we emphasize the need for long-term studies in the Neotropics, specifically in urban ecology, if we are ever to understand the causality of any spatial pattern or temporal change in urban bird ensembles and their occurrence.

Additional data would be needed to put forward some hypotheses explaining the changes in avifauna for the three cities. We are aware that climatic factors could play a role, as has been explored by Stiles et al. (2017) (that we mentioned in the discussion thanks to Reviewer #1 suggestion), with no clear conclusion for one of the cities. Nevertheless, we also included in the discussion that landscape variables could be just as relevant (such as urbanization degree in neotropical cities (i.e., Amaya-Espinel et al. 2019 in Santiago-Chile, Carvajal-Castro et al. 2019 in Armenia-Colombia, or Villegas & Zabala 2010 in La Paz-Bolivia). See lines 620 and the followings. However, we lack the appropriate data to integrate all of these parameters into a clear, testable hypothesis, calling for long-term studies to enhance their data recording.

My suggestion is that the authors think about why the differences between the regions occur and try to use GLMs/GLMMs to explain the differences using local covariates (the urbanization degree, for example).

As we pointed out in the previous answer, this study seeks to highlight first how long-term monitoring studies allow us to recognize the spatial and temporal patterns that urban bird communities have with greater precision and representativeness. In this case, using the GLMs allowed us to approach detecting both differences between the counting sites in the cities and recognizing intra-annual variations in each of these places. GLMs can help establish some urban covariates' contribution to explaining these patterns. However, to date, there has yet to be any information available on the three cities in Colombia to develop that analysis, mainly due to the temporal variation that the information may have (for example, green or gray covers, local vegetation, climate). Advancing in this analysis implies first having time series of these covariates, as well as possibly venturing into multitemporal models, for example, of occurrence that, in addition to addressing variables that define presence or absence, deal with variables that affect species detectability or even with parameters such as colonization and extinction. In addition, those models operate at the species-specific level. This was not the final purpose of this study, but we consider the Reviewer's comment to be very pertinent. We have therefore included the relevance of moving forward in this direction for Colombian cities and other cities in the Neotropical region that have been developing long-term monitoring of birds. See lines 620 as well as 704 and the following. 

Also, the authors could explore the differences among years in the number of species recorded.

Among the general results in the first paragraphs of the result section (lines 284 and so on), we included references to the most common species and families in each city, with some yearly variations. However, identifying the reviewers' recommendation, we have included some other yearly differences in species that are primarily sporadic (few counts) along the 18 years per city.

And I highly recommend using models that include estimation of parameters corrected by imperfect detection (hierarchical community models, for example).

We appreciate the suggestion, which we considered during the data analysis stage. Unfortunately, as mentioned previously, there is no information on the three cities to develop hierarchical community models, especially due to the lack of information on the one hand about covariates that can affect the probability of occurrence of the bird species (for example, green or gray covers, vegetation composition, green area size) or in its detectability (climate conditions, number of observers, tree cover, etc.). Unfortunately, these last variables that the observers can complete in the count formats of each place visited each year are usually omitted and represent a critical gap that we included in the discussion (see lines 599 and 664).

Figure 1: The color of the different departments is orange and it’s not possible to distinguish which is each one.

The authors need to explain the meaning of the red dots in the maps. Is it the sampling locations?

As was noted with the comment of Reviewer #1, we changed to more contrasting colors and have included the meaning of the dots in the figure's legend.

Figure 2: Needs improvement. The authors should add a legend in figure showing what each color mean, even if it’s in the figure description. Also, the sites’ names should be in the top of each panel to make the interpretation easier. The authors need to take into account that some readers will only take a look at the figures, so they need to be as clear as possible.

We have included the Reviewer's suggestions in the figure and adjusted the figure legend accordingly.

Figure 3: Needs more information. The acronyms of each site should be explained in the figure description. Add the name of the cities in the top of each panel.

We have included the Reviewer's suggestions in the figure and adjusted the figure legend, also calling the reader's attention to Table in Supporting material 1 for extensive detail.

Figure 4: Hard to read because the axis font is too small. Needs improvement.

We have enlarged the font and included other information in the legend to make it more comprehensible.

The ‘Study area’ topic fits better if inside the overall ‘Methods’ topic.

We have followed such a suggestion and included it in the methods section.

Lines 361-366: Again, abundance is an estimated parameter. The authors are referring to counts, not abundance in this paragraph.

We thank the Reviewer for such clarification and have followed the idea throughout the manuscript and the figures.

It’s hard to compare raw counts from places with different environmental characteristics.

Indeed, as such, and as we have stated previously, this manuscript does not aim to perform such comparisons.

Table 1: Really hard to understand. The authors need to organize better this table. Does dividing the information by site really matters? What does the letters in the side of the numbers mean? Consider add this table in the supplemental material.

All tables accompanying the manuscript are in the supporting material. Therefore we consider the comment relates to Table 1 S1.

Following the suggestion by Reviewer #1, we have clarified the Table heading. We include all this information, which is not recorded in the open-access databases, for other researchers or colleagues to evaluate the results in light of the metadata presented therein. As we mentioned in the manuscript, not all metadata is complete for every year, at every site in every city, but we feel having this information could enable more exploration on data analysis.

Line 445: Better not start a paragraph with ‘However’.

We thank the reviewer and have adjusted the grammatical suggestion.

Figure 5: Add the name of the cities and the urban category description in the figure too. The acronyms make it harder to understand.

We have included the Reviewer's suggestions in the figure and adjusted the figure legend accordingly to avoid redundancy.

---

## [Decision Letter · Decision Letter 1]

17 Jan 2023

Long-term Christmas Bird Counts describe Neotropical urban bird diversity

PONE-D-22-20926R1

Dear Dr. Galvis,

We’re pleased to inform you that your manuscript has been judged scientifically suitable for publication and will be formally accepted for publication once it meets all outstanding technical requirements.

Kind regards,

Daniel de Paiva Silva, Ph.D.

Academic Editor

PLOS ONE

Additional Editor Comments (optional):

Dear Dr. Galvis,

I am pleased to inform youthat your manuscript has been accepted for publication in PLoS One! There is a minor improvement to be made but it can be handled in the proofreading stage.

Sincerely,

Daniel Silva

Reviewers' comments:

Reviewer's Responses to Questions

**Comments to the Author**

1. If the authors have adequately addressed your comments raised in a previous round of review and you feel that this manuscript is now acceptable for publication, you may indicate that here to bypass the “Comments to the Author” section, enter your conflict of interest statement in the “Confidential to Editor” section, and submit your "Accept" recommendation.

Reviewer #1: All comments have been addressed

Reviewer #2: All comments have been addressed

2. Is the manuscript technically sound, and do the data support the conclusions?

Reviewer #1: Yes

Reviewer #2: Yes

3. Has the statistical analysis been performed appropriately and rigorously? 

Reviewer #1: Yes

Reviewer #2: Yes

4. Have the authors made all data underlying the findings in their manuscript fully available?

Reviewer #1: Yes

Reviewer #2: Yes

5. Is the manuscript presented in an intelligible fashion and written in standard English?

Reviewer #1: Yes

Reviewer #2: Yes

6. Review Comments to the Author

Reviewer #1: (No Response)

Reviewer #2: The authors improved the manuscript based on the reviewers comments and, I my opinion, the manuscript is suitable for publication.

The minor comment is:

Line 283: Change for "...the most recorded species..."

7. PLOS authors have the option to publish the peer review history of their article (what does this mean?). If published, this will include your full peer review and any attached files.

Reviewer #1: No

Reviewer #2: No

---

## [Editor Report · Acceptance letter]

23 Jan 2023

PONE-D-22-20926R1 

Long-term Christmas Bird Counts describe Neotropical urban bird diversity 

Dear Dr. Echeverry-Galvis:

I'm pleased to inform you that your manuscript has been deemed suitable for publication in PLOS ONE. Congratulations! Your manuscript is now with our production department. 

Kind regards, 

on behalf of

Dr. Daniel de Paiva Silva 

Academic Editor

PLOS ONE